# Robust disease prognosis via diagnostic knowledge preservation: A sequential learning approach

Haresh Rengaraj Rajamohan[1]*, Yanqi Xu[1], Weicheng Zhu[1], Richard Kijowski[2], Kyunghyun Cho[1], Krzysztof J. Geras[1,3], Narges Razavian[3,4], Cem M. Deniz[3,5]

1 Center for Data Science, New York University, New York, New York, United States of America, 2 Department of Radiology, Hospital for Special Surgery, New York, New York, United States of America, 3 Department of Radiology, New York University Langone Health, New York, New York, United States of America, 4 Department of Population Health, New York University Langone Health, New York, New York, United States of America, 5 Bernard and Irene Schwartz Center for Biomedical Imaging, New York University Langone Health, New York, New York, United States of America

* hrr288@nyu.edu

## Abstract

Accurate disease prognosis is essential for patient care but is often hindered by the scarcity of longitudinal data. This study explores deep learning training strategies that utilize large, accessible diagnostic datasets to pretrain models aimed at predicting future disease progression in knee osteoarthritis (OA), Alzheimer's disease (AD), and breast cancer (BC). While diagnostic pretraining improves prognostic task performance, naive fine-tuning for prognosis can cause 'catastrophic forgetting,' where the model's original diagnostic accuracy degrades, a significant patient safety concern in real-world settings. To address this, we propose a sequential learning strategy with experience replay. We used cohorts with knee radiographs, brain MRIs, and digital mammograms to predict 4-year structural worsening in OA, 2-year cognitive decline in AD, and 5-year cancer diagnosis in BC. Our results showed that diagnostic pretraining on larger datasets improved prognosis model performance compared to standard baselines, boosting both the Area Under the Receiver Operating Characteristic curve (AUROC) (e.g., Knee OA external: 0.770 vs 0.747; Breast Cancer: 0.874 vs 0.848) and the Area Under the Precision-Recall Curve (AUPRC) (e.g., Alzheimer's Disease: 0.752 vs 0.683). Additionally, a sequential learning approach with experience replay achieved prognostic performance comparable to dedicated single-task models (e.g., Breast Cancer AUROC 0.876 vs 0.874) while also preserving diagnostic ability. This method maintained high diagnostic accuracy (e.g., Breast Cancer Balanced Accuracy 50.4% vs 50.9% for a dedicated diagnostic model), unlike simpler multitask methods prone to catastrophic forgetting (e.g., 37.7%). Our findings show that leveraging large diagnostic datasets is a reliable and data-efficient way to enhance prognostic models while maintaining essential diagnostic skills.

**Data availability statement:** - Osteoarthritis Initiative (OAI): OAI data are publicly available through the NIMH Data Archive (NDA). Access requires creating an NDA account, agreeing to the OAI data access terms, and requesting the desired collections through the NDA portal. Full instructions are provided at https://nda.nih.gov/oai. - Multicenter Osteoarthritis Study (MOST): MOST data are publicly available through the NIA Aging Research Biobank. Access requires creating a Biobank account and submitting a data request in accordance with the Biobank's terms and conditions. More information is available at https://agingresearchbiobank.nia.nih.gov/. - Alzheimer's Disease Neuroimaging Initiative (ADNI): ADNI data are publicly available from the LONI Image & Data Archive upon completion of a web application and acceptance of the ADNI Data Use Agreement. Data can be accessed at https://adni.loni.usc.edu/. - Breast Cancer Cohort (NYU Langone Health): This institutional dataset contains protected health information and cannot be made publicly available. De-identified data may be shared upon reasonable request, pending approval by NYU Langone's data governance committee. Inquiries can be directed to Krzysztof Geras (k.j.geras@nyu.edu). - Code and Model Availability: All code for preprocessing and model training sufficient to reproduce the analyses reported in this study are available at https://github.com/denizlab/diag-to-prog-replay.

**Funding:** This study was supported by the National Institute of Arthritis and Musculoskeletal and Skin Diseases in the form of a grant awarded to CMD (R01-AR074453) and the National Institute of Arthritis and Musculoskeletal and Skin Diseases in the form of a salary for HRR. The specific roles of these authors are articulated in the 'author contributions' section. The funders had no role in study design, data collection and analysis, decision to publish, or preparation of the manuscript.

**Competing interests:** The authors have declared that no competing interests exist.

## 1. Introduction

Progressive diseases present significant challenges to healthcare systems worldwide. Their gradual onset and potential for irreversible damage make early intervention, guided by accurate prognosis, crucial for optimizing patient outcomes, reducing healthcare costs, and improving quality of life [1–3]. Key examples include knee osteoarthritis (OA), Alzheimer's disease (AD), and breast cancer (BC) – conditions representing diverse physiological systems but sharing the critical need for early identification and reliable progression prediction [1,3–5].

Deep learning (DL) [6] offers powerful tools for automating medical image analysis and has shown considerable promise for both diagnosis and prognosis tasks across various diseases. Studies have demonstrated DL's effectiveness in diagnosing OA severity [7], detecting AD from neuroimaging [8–11], and screening for BC [12,13]. Similarly, DL models have been developed to predict disease progression, such as total knee replacement risk in OA [14–18], progression to AD [19,20], and long-term BC outcomes [21,22].

However, developing robust prognosis models faces a significant hurdle: the scarcity of longitudinal data required to track disease progression over time. This contrasts with diagnosis, where larger cross-sectional datasets are often more readily available. As a result, recent advances in medical imaging AI – across knee osteoarthritis, Alzheimer's disease, and oncology have largely focused on improving diagnostic accuracy and disease severity assessment, reflecting the scale and availability of diagnostic datasets rather than long-horizon progression labels [23–25]. To mitigate this data scarcity, previous works have explored multitask learning (MTL) as additional regularization, where a single model is trained to perform several related tasks simultaneously, aiming to improve generalization and performance through shared representations [13,14,16,21,22]. For instance, MTL combining diagnosis and prognosis has shown benefits in OA [14,16], and pretraining on BI-RADS has been shown to improve current cancer diagnosis performance [13]. Yet these MTL studies typically rely on relatively small prognosis datasets and do not fully capitalize on the abundance of diagnostic data, even as recent medical foundation-model approaches increasingly leverage large-scale diagnostic pretraining with limited labeled data [26,27]. This study therefore hypothesizes that sequential learning – diagnosis pretraining on large, unbiased datasets followed by prognostic training, achieves a favorable balance of prognostic and diagnostic performance.

A key challenge arises, however, when building a single model that learns tasks sequentially. When a model pretrained on a data-rich task (e.g., diagnosis) is subsequently fine-tuned on a new, data-scarce task (e.g., prognosis), it often suffers from catastrophic forgetting, a significant degradation of its ability to perform the original task [28]. In the context of healthcare, this phenomenon is not merely a technical limitation but a critical issue for real-world model maintenance and patient safety. A prognostic model that compromises diagnostic accuracy could lead to missed disease during routine follow-up, undermining its clinical utility and trust. A primary strategy to combat this phenomenon is experience replay (or rehearsal), where the model is periodically retrained on stored examples from the original task while learning the new one, thereby refreshing and preserving its previously acquired knowledge [29].

This work demonstrates that pretraining models on larger diagnostic datasets significantly improves prognostic performance, generalization to external cohorts, and discrimination within patient subgroups compared to standard initialization techniques across knee OA, Alzheimer's disease and breast cancer. We also identify that a sequential multitask learning approach with experience replay yields the best overall results, matching the prognostic performance of dedicated single-task models while effectively preserving diagnostic capabilities and outperforming simpler joint training strategies.

## 2. Materials and methods

### 2.1. Ethics statement

This retrospective study involved the analysis of fully de-identified data from three publicly available research datasets (OAI, MOST, and ADNI) and one institutional dataset (NYU Langone Health). The **Osteoarthritis Initiative (OAI)** (ClinicalTrials.gov identifier: NCT00080171) was approved by the Institutional Review Boards (IRB) at the UCSF Coordinating Center (approval #10–00532) and all clinical sites, including Memorial Hospital of Rhode Island, Ohio State University, University of Pittsburgh, and University of Maryland/Johns Hopkins University. Ethical approval for the **Multicenter Osteoarthritis Study (MOST)** was obtained from the IRBs at Boston University (H-32956), University of Alabama at Birmingham (IRB-000329007), University of California San Francisco (301480), and University of Iowa (201511711); approval for secondary data analysis was granted by the University of Florida (IRB202201899). Data for the **Alzheimer's Disease Neuroimaging Initiative (ADNI** ClinicalTrials.gov identifier: NCT00106899)** were obtained in accordance with the Declaration of Helsinki and approved by the IRBs of all participating institutions. All participants in these public cohorts provided written informed consent. The **Breast Cancer** dataset was sourced from New York University Langone Health under a protocol approved by the NYU Langone Health IRB (IRB00010481, S18-00712). This dataset consisted of fully de-identified data and was approved for retrospective analysis with a waiver of informed consent, in compliance with HIPAA regulations. As this study involved only the retrospective analysis of fully de-identified data, it did not require new participant recruitment or additional IRB approval beyond the existing protocols cited above.

### 2.2. Prediction task definitions

We investigate the following clinical measures for knee OA, Alzheimer's disease and breast cancer:

#### 2.2.1. Knee Osteoarthritis (OA).

**Diagnosis:** Classification based on Kellgren-Lawrence Grade (KLG) [30] – A severity scale (0–4) assigned by radiologists based on radiographic features including joint space narrowing, bone spurs, and sclerosis.

**Prognosis:** Prediction of disease worsening over a 4-year horizon [31], defined as:

- Structural Incidence for early-stage OA (KLG 0–1): Progression to KLG ≥ 2 or undergoing a Total Knee Replacement (TKR).

- Structural Progression for radiographic OA (KLG ≥ 2): An increase in KLG or undergoing a TKR.

#### 2.2.2. Alzheimer's Disease (AD).

**Diagnosis:** Classification of cognitive status using structural brain MRI, based on the diagnostic criteria established by the **Alzheimer's Disease Neuroimaging Initiative (**ADNI) study protocol:

- Cognitive Normal (CN): Participants with no significant impairment in cognitive functions or activities of daily living.

- Mild Cognitive Impairment (MCI): Participants with a subjective memory concern, objective memory loss, and preserved general cognition and functional performance.

- Alzheimer's Disease (AD): Participants meeting the NINCDS/ADRDA criteria for probable AD.

**Prognosis:** Prediction of cognitive decline over a 2-year horizon:

- CN → MCI: Onset of cognitive impairment.

- MCI→AD or CN→AD: Progression to confirmed disease.

**2.2.3. Breast Cancer (BC). BI-RADS (Diagnosis):** Classification system for breast imaging findings:

- Grade 0: Incomplete assessment; additional imaging evaluation is needed.

- Grade 1: Negative; no abnormalities found.

- Grade 2: Benign finding; no malignant characteristics present.

**Prognosis:** Prediction of a patient having a confirmed breast cancer diagnosis within 5 years of the scan date.

## 2.3. Datasets and cohorts

**2.3.1. Knee osteoarthritis.** Data were sourced from the **Osteoarthritis Initiative (OAI)** [32] for model development and the **Multicenter Osteoarthritis Study (MOST)** [33] for external validation. The OAI is a long-term, multicenter observational study of 4,796 participants focused on knee OA biomarkers. We utilized bilateral posterior-anterior fixed flexion knee radiographs.

- **Diagnosis Cohort (OAI):** Comprised 47,041 radiographs from 4,508 subjects with KLG assessments.

- **Prognosis Cohort (OAI):** To predict structural worsening, a 4 year follow up from the baseline visit was used. This timeframe was selected to balance observing meaningful change and maximizing cohort size, as longer horizons led to significant participant drop-off. Further, knees with KLG 4 or a TKR at baseline were excluded. Two distinct prognosis tasks and their corresponding cohorts were defined based on baseline disease severity:

  - **Structural Incidence:** This task focused on predicting the onset of radiographic OA in knees with baseline KLG 0–1. The cohort included 420 patients who progressed (defined as reaching KLG ≥ 2 or undergoing a TKR) and 2,439 non-progressing controls.

  - **Structural Progression:** This task focused on predicting the worsening of established OA in knees with baseline KLG 2–3. The cohort included 571 patients who progressed (defined as an increase in KLG or undergoing a TKR) and 1,677 non-progressing controls.

- **MOST Validation Cohort:** Corresponding cohorts were created from the MOST dataset using a 5-year follow-up due to data availability. The incidence cohort included 505 progressors and 1,395 controls. The progression cohort included 562 progressors and 617 controls.

**2.3.2. Alzheimer's disease.** Data were obtained from the ADNI database [25], a longitudinal multicenter study aimed at validating biomarkers for AD progression. We utilized T1-weighted brain MRI scans.

- **Diagnosis Cohort:** Consisted of 2,723 scans from 662 patients classified as CN, MCI or AD.

- **Prognosis Cohort:** To predict cognitive decline within a 24-month follow-up, 914 scans from 365 unique patients were used. Progression was defined as a change from CN to MCI/AD or MCI to AD. To augment the dataset size, MRI scans from both baseline and available intermediate follow-up visits were utilized as distinct input time points; however, scans from patients already classified as AD were excluded. To prevent data leakage and ensure the model learned generalizable prognostic features rather than patient-specific anatomy, all scans from a single patient were strictly kept within the same data partition (training, validation, or test) during all experiments. The cohort included 148 progressing patients (contributing 343 scans) and 217 stable controls (contributing 571 scans).

**2.3.3. Breast cancer.** We utilized a curated dataset of digital screening mammograms from NYU Langone Health [34], enhanced with longitudinal follow-up cancer labels up to five years post-imaging.

- **Diagnosis Cohort:** Contained 56,733 examinations from 33,681 patients with radiologist-assigned BI-RADS grades 0, 1, or 2.

- **Prognosis Cohort:** To facilitate robust model development, a balanced cohort was constructed. This included 3,000 examinations from 2,319 patients who were diagnosed with breast cancer within 5 years (but more than 130 days after the index scan) and 3,000 control examinations from 2,837 patients who remained cancer-free for the 5-year follow-up period. The mean age for patients contributing case examinations was 61.5±11.4 years, while the mean age for controls was 56.4±10.8 years.

To prevent data leakage, strict patient-level splits were used to ensure disjoint sets of individuals across all training, validation, and test sets, and no information from future visits was used as input features. Detailed demographic and clinical characteristics for all patient cohorts are provided in S1-S4 Tables.

### 2.4. Approach

Our work investigates two key approaches: utilizing diagnosis tasks for model pretraining and proposing an effective multitask learning approach.

**2.4.1. Diagnosis pretraining.** We compare diagnosis-based initialization against modality-specific baselines (Fig 1). We use modality-appropriate initializations: OA is 2D and moderate-resolution, where ImageNet pretraining is standard; AD uses 3D brain MRI (no ImageNet analogue), and BC requires very high-resolution mammography where running large ImageNet backbones at native resolution is not tractable, so we use random initialization similar to previous works [8,12]. Concretely, prognosis models are initialized from either (i) diagnosis-pretrained weights learned on the corresponding diagnostic task, (ii) **ImageNet weights** (OA only), or (iii) **random weights** (AD, BC).

**2.4.2. Sequential learning with experience replay.** To mitigate catastrophic forgetting [28,29] in multitask settings, we used a two-phase approach (Fig 1).

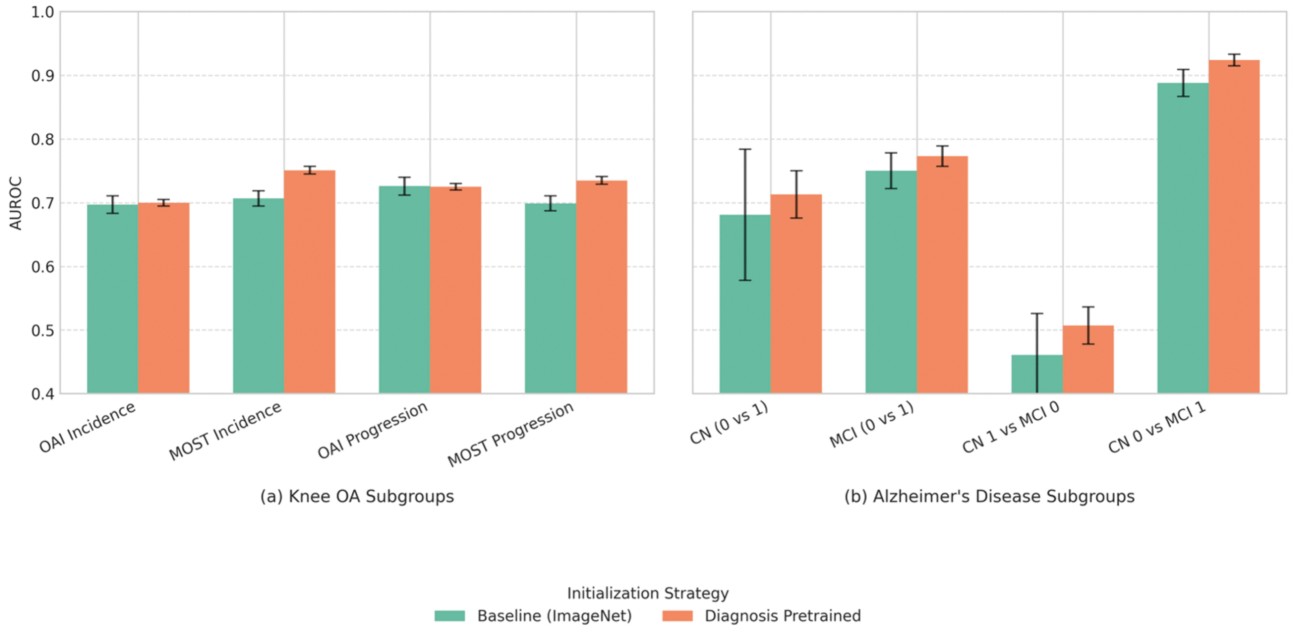

**Fig 1. Illustrates the proposed approaches (a) single task prognosis with diagnostic pretraining and (b) Sequential learning with experience replay.**

1. **Phase 1 (Diagnosis Pretraining):** A model is trained on the large diagnosis cohort.

2. **Phase 2 (Prognosis training with Diagnosis Replay):** The model is fine-tuned using a multitask objective. At each step, a batch is randomly sampled with equal probability from either the prognosis cohort (to update on the prognosis task) or the diagnosis cohort (to "replay" and refresh diagnosis knowledge). This allows the model to learn the new task while preserving its original capabilities.

### 2.4.3. Comparative multitask approaches.

For comparison, we evaluated three alternative strategies:

- **Single-cohort MT:** An approach similar to that used by Tiulpin et al. [14] and Leung et al. [16], where the model is trained on both diagnosis and prognosis tasks using only the data available in the smaller prognosis cohort.

- **Diagnosis-pretrained MT:** Initializes with diagnostic pretraining but then fine-tunes solely on the prognosis cohort's diagnosis and prognosis labels, without experience replay

- **Concurrent MT:** Trains simultaneously on both diagnosis and prognosis cohorts from a baseline initialization, without any pretraining.

## 2.5. Experimental setup

### 2.5.1. Model architectures and preprocessing.

- **OA:** Input radiographs were normalized, and knee joints were extracted prior to analysis. ResNet-34 with attention [35,36] architecture was used on extracted knee joints.

- **AD:** Input MRI scans underwent standard bias correction and spatial normalization. A custom 3D-CNN [8] was used on processed T1 MRIs.

- **BC:** The four standard mammographic views were individually processed and cropped to a uniform resolution. A custom multi-view CNN [12] was used on the processed mammogram views.

### 2.5.2. Training protocol.

All models were trained using the Adam optimizer [37] coupled with a cosine annealing learning rate scheduler to ensure stable convergence. For the highly imbalanced breast cancer BI-RADS task, a weighted cross-entropy loss was employed to give appropriate importance to rare classes. Condition-specific data augmentation techniques (including random cropping, flips, and rotations) were applied during training to enhance model robustness.

For tasks with smaller sample sizes (all prognosis cohorts and the AD diagnosis cohort), we employed a 5-fold cross-validation scheme with strict, non-overlapping patient-level splits. For each of the 5 iterations, one fold was designated as the held-out test set. Of the remaining four folds, three were used for training the model, and one was used as the validation set for hyperparameter tuning and model checkpointing via early stopping. This process was repeated five times, ensuring that every patient served as part of a test set exactly once. The final reported performance metrics are the mean and standard deviation across the results from the five test folds.

For multitask learning with experience replay, tasks were sampled with equal probability (p = 0.5). Crucially, to prioritize the more challenging prognostic task, model checkpoints were saved based on the epoch achieving the highest prognosis AUROC on the validation set, rather than on the total loss

For each disease, the diagnosis and prognosis cohorts were drawn from the same underlying patient populations (OAI, ADNI, and NYU dataset). As such, patients in the longitudinal prognosis cohort were also represented in the larger cross-sectional diagnosis cohort. To prevent any data leakage between the pretraining and fine-tuning stages, a strict patient-level splitting procedure was enforced. For each of the 5 cross-validation folds of the prognosis task, the patient

IDs assigned to the test set were identified first. These testset patients were then completely excluded from the training set of the large diagnosis model used for pretraining. This 'test set holdout' strategy ensures that our prognosis models were always evaluated on patients that the entire training pipeline, including the pretraining stage, had never seen, providing an unbiased estimate of performance.

### 2.5.3. Evaluation and analysis.

This study was prepared and reported in accordance with the TRIPOD (Transparent Reporting of a multivariable prediction model for Individual Prognosis or Diagnosis) guideline wherever applicable. The pre-specified primary endpoint for all prognostic tasks was the Area Under the Receiver Operating Characteristic curve (AUROC), which measures the model's overall ability to discriminate between patients who experience disease progression and those who do not.

Secondary endpoints for prognosis included the Area Under the Precision-Recall Curve (AUPRC), which provides additional insight into model performance, especially in cohorts with class imbalance. Sensitivity and specificity were computed additionally for breast cancer.

For the threshold-dependent metrics of sensitivity and specificity, the following procedure was used for each of the 5 cross-validation folds: first, an optimal decision threshold was determined by maximizing the Youden Index [38] on the validation set for that fold. This fold-specific threshold was then applied to the predictions on the corresponding test set to calculate sensitivity and specificity. The final reported metrics are the mean and standard deviation of the five resulting sensitivity and specificity values.

For diagnosis, we report balanced accuracy for the five-class knee OA task, following [16] to mitigate class imbalance. For the multi-class AD and BC tasks, we additionally report macro-AUROC and micro-AUROC, consistent with prior work. Macro-AUROC is the unweighted mean of one-vs-rest AUROCs across classes, giving each class equal weight. Micro-AUROC pools predictions and labels across all classes to compute a single AUROC, effectively weighting classes by their prevalence. Balanced accuracy is the mean of per-class recall (sensitivity).

To formally compare the primary approaches, ROC curves of ensembled cross-validation predictions were compared using the DeLong test [39]. For every patient in the test set, the predictive scores (probabilities) from the five respective cross-validation models were averaged. This resulted in a single set of ensembled predictions for the entire test cohort, from which a single, more stable ROC curve was generated for the statistical comparison. This practice of averaging predictions is a standard ensembling technique to reduce variance and provide a more robust estimate of model performance.

Stratified subgroup analyses were performed on pre-defined clinical subgroups: structural incidence (KL 0–1) vs. progression (KL 2–3) for Knee OA, and cognitive transition groups (e.g., CN vs. MCI) for Alzheimer's Disease. These subgroup analyses were conducted to verify that the models were learning genuine prognostic signals rather than simply using baseline disease severity as a proxy for progression risk. This analysis was not powered for formal statistical tests of subgroup interaction. Statistical significance testing was focused on our primary hypothesis comparing the baseline and diagnosis-pretrained models.

## 3. Results

The experimental results are presented in two main parts. The first subsection evaluates the impact of initializing prognosis models with weights pretrained on large diagnostic datasets, comparing their performance against standard baselines for single-task prognosis. The second subsection compares four multitask learning strategies on both prognosis and diagnosis.

### 3.1. Impact of diagnosis pretraining

Initializing prognosis models with weights pretrained on diagnostic tasks with full diagnosis cohorts yielded improvements over baseline initializations (ImageNet for Knee OA, random for AD/BC), as detailed in Table 1.

For Knee Osteoarthritis, while performance on the internal OAI dataset was comparable, diagnosis-pretrained models demonstrated superior generalization on the external MOST dataset (e.g., AUROC 0.770 vs 0.747) and exhibited lower variance, suggesting more stable performance. This improved generalization on the external MOST dataset was

**Table 1. Comparison of prognosis prediction performance using models with baseline initialization versus diagnostic pretraining across three diseases. Knee OA models used ImageNet pretraining as baseline; AD and Breast Cancer models used random initialization as baseline.**

| Disease | Dataset | Metric | Baseline Init. | Diagnosis Pretrained Init. |
|---|---|---|---|---|
| Knee Osteoarthritis | OAI (Internal) | AUROC | **0.744±0.014** | 0.738±0.005 |
| | | AUPRC | **0.452±0.014** | 0.448±0.008 |
| | MOST (External) | AUROC | 0.747±0.012 | **0.770±0.006** |
| | | AUPRC | 0.564±0.011 | **0.602±0.008** |
| Alzheimer's Disease | ADNI | AUROC | 0.807±0.025 | **0.840±0.008** |
| | | AUPRC | 0.683±0.030 | **0.752±0.012** |
| | | Sensitivity | 60.5%±12.0% | **78.1%±8.2%** |
| | | Specificity | **82.5%±6.1%** | 72.6%±11.8% |
| Breast Cancer | NYU | AUROC | 0.848±0.007 | **0.874±0.002** |
| | | AUPRC | 0.873±0.012 | **0.900±0.001** |
| | | Sensitivity | 67.6%±2.1% | **69.9%±1.8%** |
| | | Specificity | 92.0%±3.5% | **93.4%±2.7%** |

*Note: AUROC: Area Under the Receiver Operating Characteristic curve; AUPRC: Area Under the Precision-Recall curve. Values are mean±standard deviation. Boldface indicates the best-performing approach for each metric.

statistically significant when comparing model ensembles via DeLong's test (AUROC 0.781 for diagnosis-pretrained vs. 0.768 for ImageNet-pretrained, $p = 0.006$, n = 4251). Additionally, this performance benefit on the external MOST dataset was consistent across both early-stage (Incidence) and established (Progression) disease subgroups (Fig 2a).

For Alzheimer's Disease, diagnostic pretraining significantly enhanced AD progression prediction across all metrics, notably increasing AUROC (0.840 vs 0.807) and AUPRC (0.752 vs 0.683). Benefits were observed across different cognitive subgroups (Fig 2b). However, the difference in overall AUROC between model ensembles on the whole test set was not statistically significant via DeLong's test (0.846 for diagnosis-pretrained vs. 0.823 for random initialized, $p = 0.137$, n = 236).

For Breast Cancer, diagnostic pretraining substantially improved 5-year prognosis prediction (AUROC 0.874 vs 0.848). This boost extended to precision-recall (AUPRC 0.900 vs 0.873) and clinical utility metrics (sensitivity/specificity), alongside improved stability (lower std. dev.). This performance increase was statistically significant when comparing model ensembles (AUROC 0.876 for diagnostic-pretrained vs. 0.854 for random initialized, $p < 0.001$, n = 1735).

Across all three diseases, these findings reinforce that leveraging diagnostic knowledge provides a valuable initialization for prognostic tasks, likely by helping the model learn representations of relevant anatomical and pathological features that precede clinically evident progression.

### 3.2. Multitask learning performance

Four multitask learning strategies were further compared, evaluating performance on both the prognosis task (Table 2) and the diagnosis task (Table 3).

For Knee Osteoarthritis, Sequential Learning with Experience Replay and Diagnosis-pretrained MT demonstrated the strongest prognostic performance (e.g., OAI AUROC 0.75, MOST AUROC 0.764), matching the dedicated single-task reference model. On the diagnosis task, Sequential Learning with Experience Replay maintained robust performance (OAI Bal Acc 75.1%), while other methods showed substantial degradation.

For Alzheimer's Disease, Sequential Learning with Experience Replay and Diagnosis-pretrained MT again achieved the highest prognosis performance (AUROC 0.851 and 0.848). For diagnosis, Sequential Learning with Experience Replay preserved strong performance (macro AUROC 0.758), whereas Single-cohort MT and Diagnosis-pretrained MT suffered significant degradation (macro AUROC 0.53).

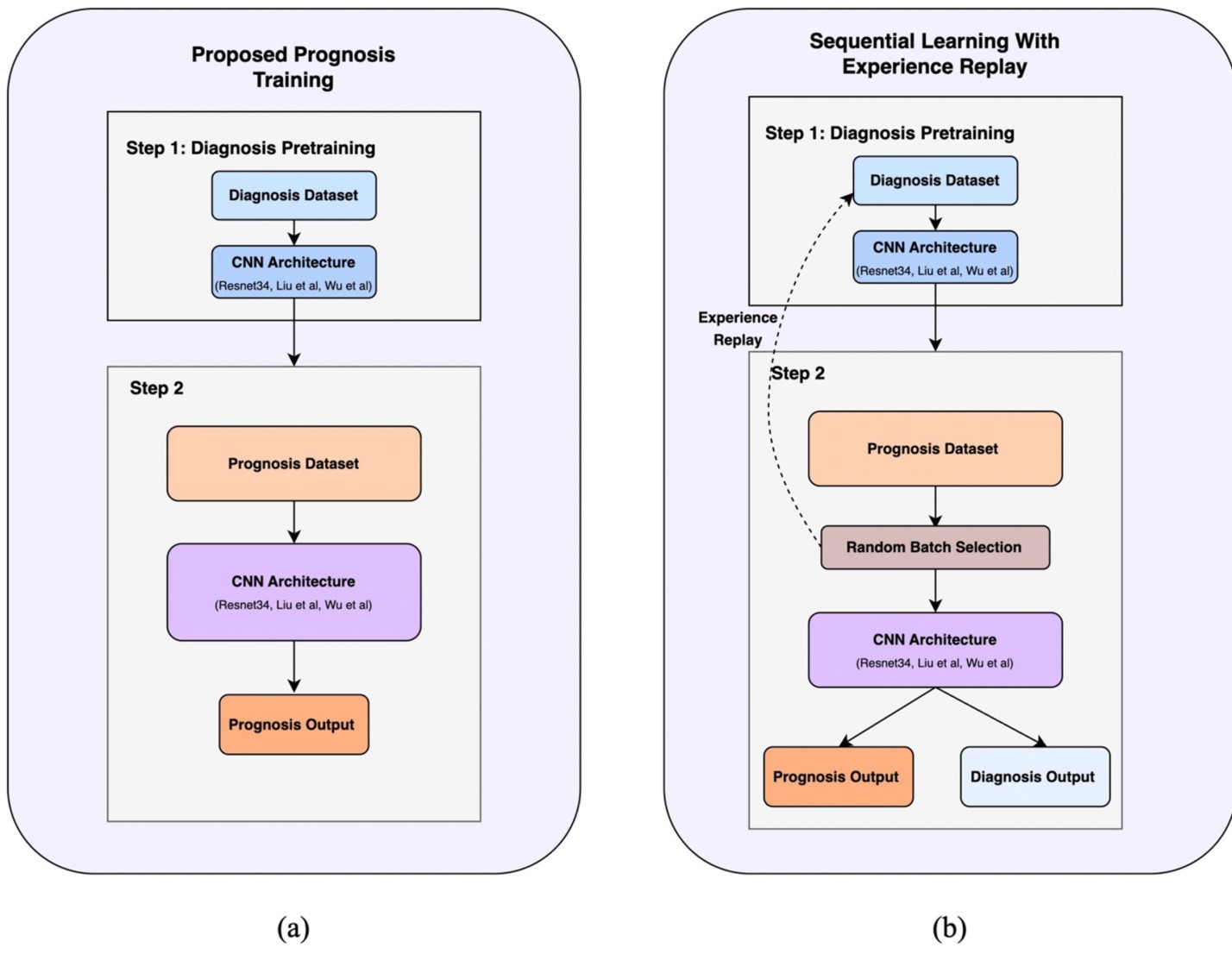

**Fig 2. Comparison of baseline initialization versus diagnostic pretraining for prognosis prediction performance (AUROC) within specific disease subgroups. (a)** Knee OA subgroups include Incidence (progression from KL 0-1) and Progression (worsening from KL 2-3) cohorts evaluated on the internal OAI and external MOST datasets. **(b)** Alzheimer's Disease subgroups represent different cognitive transitions within the ADNI dataset (CN: Cognitively Normal, MCI: Mild Cognitive Impairment; 0: Stable status over 2 years, 1: Progressing status over 2 years). Diagnostic pretraining generally improves or maintains performance with better stability, showing particular benefit in the external MOST and all AD sub cohorts. Full numerical results are available in S5 and S6 Tables.

For Breast Cancer, Sequential Learning with Experience Replay and Diagnosis-pretrained MT yielded the best prognosis performance (AUROC 0.876 and 0.878 respectively). Critically, Sequential Learning with Experience Replay maintained diagnostic metrics nearly identical to the single-task reference model (Bal Acc 50.4% vs 50.9%), while all other multitask methods showed poor to degraded diagnostic performance.

The robustness of the Sequential Learning with Experience Replay approach was further confirmed in the subgroup analyses (Fig 3). Across the various Knee OA and Alzheimer's Disease subgroups, this method consistently achieved prognostic performance that was comparable or superior to the dedicated single-task reference model. In contrast, simpler

**Table 2. Prognosis prediction performance across different multitask learning approaches.** 'Ref (Diag. Pret)' refers to the single-task prognosis model initialized with diagnostic pretraining (from Table 1). 'Single-cohort MT' uses only prognosis cohort data for multitask training. 'Concurrent MT' trains on diagnosis and prognosis cohorts simultaneously from baseline initialization. 'Diag-pret MT' performs diagnostic pretraining then trains multitask on the prognosis cohort. 'Seq Learn w/ Replay' initializes with diagnosis pretraining, trains multitask on prognosis cohort and uses experience replay with the diagnosis cohort.

| Disease | Dataset | Metric | Single-task prognosis (diagnosis-pretrained init.) | Single-cohort MT | Concurrent MT | Diag-pret MT | Seq Learn w/ Replay |
|---|---|---|---|---|---|---|---|
| Knee Osteoarthritis | OAI (Internal) | AUROC | 0.738±0.005 | 0.739±0.017 | 0.743±0.011 | 0.748±0.005 | **0.750±0.006** |
| | | AUPRC | 0.448±0.008 | 0.444±0.015 | 0.438±0.011 | **0.458±0.005** | 0.456±0.006 |
| | MOST (External) | AUROC | **0.770±0.006** | 0.756±0.027 | 0.765±0.005 | 0.763±0.007 | 0.764±0.007 |
| | | AUPRC | **0.602±0.008** | 0.564±0.020 | 0.589±0.004 | 0.599±0.008 | 0.594±0.014 |
| Alzheimer's Disease | ADNI | AUROC | 0.840±0.008 | 0.798±0.014 | 0.828±0.016 | 0.848±0.008 | **0.851±0.009** |
| | | AUPRC | 0.752±0.012 | 0.653±0.027 | 0.716±0.021 | **0.754±0.011** | **0.754±0.005** |
| Breast Cancer | NYU | AUROC | 0.874±0.002 | 0.864±0.002 | 0.860±0.007 | **0.878±0.003** | 0.876±0.003 |
| | | AUPRC | 0.900±0.001 | 0.894±0.002 | 0.889±0.008 | **0.903±0.003** | **0.903±0.002** |
| | | Sensitivity | **69.9%±1.8%** | 66.1%±1.6% | 67.4%±2.6% | 69.0%±2.5% | 67.7%±1.7% |
| | | Specificity | 93.4%±2.7% | 97.1%±1.0% | 94.6%±4.3% | 95.5%±1.8% | **97.4%±1.2%** |

*Note: Values are mean±standard deviation. Boldface indicates the best-performing approach for each metric.

**Table 3. Diagnosis task performance across different multitask learning approaches.** 'Reference' refers to a model trained only on the diagnosis task. Performance is measured by Balanced Accuracy (Bal Acc) for Knee OA; and Macro AUROC, Micro AUROC, and Bal Acc for Alzheimer's Disease (AD) and Breast Cancer (BC). Other approach definitions are as in Table 2.

| Disease Condition | Dataset | Metric | Reference | Single-cohort MT | Concurrent MT | Diag-pret MT | Seq Learn w/ Replay |
|---|---|---|---|---|---|---|---|
| Knee Osteoarthritis | OAI (Internal) | Bal Acc (%) | **75.1%** | 51.9%±1.3% | 73.4%±2.4% | 56.6%±0.7% | **75.1%±1.1%** |
| | MOST (External) | Bal Acc (%) | 65.6% | 47.0%±3.0% | **67.3%±1.2%** | 48.8%±1.7% | **67.5%±2.6%** |
| Alzheimer's Disease | ADNI | Macro AUROC | **0.752±0.012** | 0.530±0.011 | 0.741±0.013 | 0.532±0.026 | **0.758±0.005** |
| | | Micro AUROC | 0.760±0.016 | 0.581±0.007 | 0.746±0.007 | 0.618±0.015 | **0.763±0.011** |
| | | Bal Acc (%) | **56.3%±4.2%** | 35.3%±3.2% | 52.7%±3.0% | 38.1%±2.9% | 53.0%±2.8% |
| Breast Cancer | NYU | Macro AUROC | **0.7** | 0.577±0.007 | 0.571±0.007 | 0.657±0.004 | **0.697±0.002** |
| | | Micro AUROC | 0.724 | 0.687±0.002 | 0.655±0.013 | 0.726±0.012 | **0.735±0.012** |
| | | Bal Acc (%) | **50.9%** | 37.7%±1.2% | 41.3%±0.4% | 45.2%±0.4% | **50.4%±0.6%** |

*Note: Values are reported as mean±standard deviation where available from cross-validation, otherwise as single run results. Boldface indicates the best-performing approach for each metric.

strategies like 'Single-cohort MT' often showed a notable degradation in performance. These results highlight Sequential Learning with Experience Replay as an optimal multitask approach, effectively preventing catastrophic forgetting while achieving top-tier prognostic performance.

## 4. Discussion

### 4.1. Principal findings

The experiments demonstrate two main findings across the studied conditions: (1) diagnostic pretraining improves prognosis prediction and provides stability, and (2) a sequential learning approach with experience replay performed best among the multitask strategies evaluated.

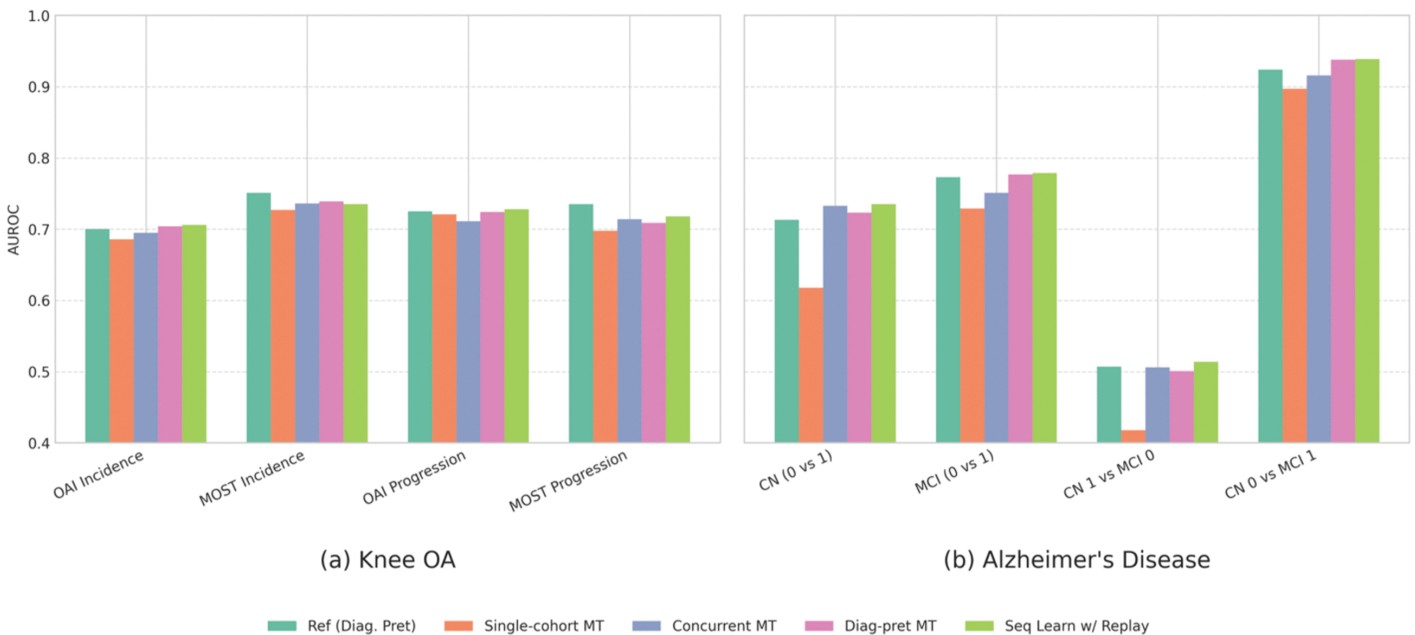

**Fig 3. Performance comparison of various multitask learning approaches for prognosis prediction (AUROC) within specific disease subgroups. (a)** Knee OA subgroups (OAI/MOST Incidence and Progression) and **(b)** Alzheimer's Disease subgroups (ADNI CN/MCI transitions) are shown. Strategies compared include a single-task reference model ('Ref (Diag. Pret)'); Multitask training using only the prognosis cohort ('Single-cohort MT'); Concurrent training on diagnosis and prognosis cohorts ('Concurrent MT'); Multitask training on the prognosis cohort after diagnostic pretraining ('Diag-pret MT'); and Sequential Learning with Experience Replay ('Seq Learn w/ Replay'). Sequential Learning with Replay demonstrates robust performance across most subgroups, often matching or exceeding the dedicated single-task reference model. Full numerical results, including standard deviations, are available in S7 and S8 Tables.

Diagnostic-pretrained initialization improved prognostic discrimination and generalization to an external cohort and across demographic/clinical subgroups. The advantage remained after controlling for diagnostic severity (Fig 2), indicating that pretraining helps models capture relevant prognostic features beyond simply using diagnosis severity as a proxy for risk.

Recent work on generalist and multimodal medical AI suggests that representations learned from large-scale diagnostic data can serve as effective foundations for a wide range of downstream diagnostic classification tasks, particularly in settings with limited labeled data [24,40]. Our results provide empirical evidence that this paradigm extends specifically to disease prognosis, where longitudinal labels are especially scarce and difficult to collect. By leveraging large, cross-sectional diagnostic cohorts, our approach improves prognostic discrimination and stability while preserving diagnostic competence, addressing a key gap between foundation-model principles and real-world longitudinal prediction tasks.

Furthermore, our evaluation of joint diagnosis-prognosis approaches showed that sequential learning with experience replay performed best. Across knee OA and AD subgroups, it reached prognostic performance comparable to dedicated, diagnosis-pretrained single-task models, while crucially, also preserving diagnostic performance. This dual success effectively mitigates the critical issue of catastrophic forgetting (Table 3, Fig 4).

In contrast, the 'Single-cohort MT' and 'Diagnosis-pretrained MT' variants were evaluated to represent simpler strategies from prior works and serve as a crucial baseline. Their performance starkly illustrates the severity of catastrophic forgetting. Lacking access to the full, unbiased diagnostic dataset during fine-tuning, these approaches suffered a sharp degradation in diagnostic ability. They effectively forgot knowledge from pretraining, including how to classify advanced disease stages in knee OA/AD and the rare class BI-RADS 0 in breast cancer (Fig 4).

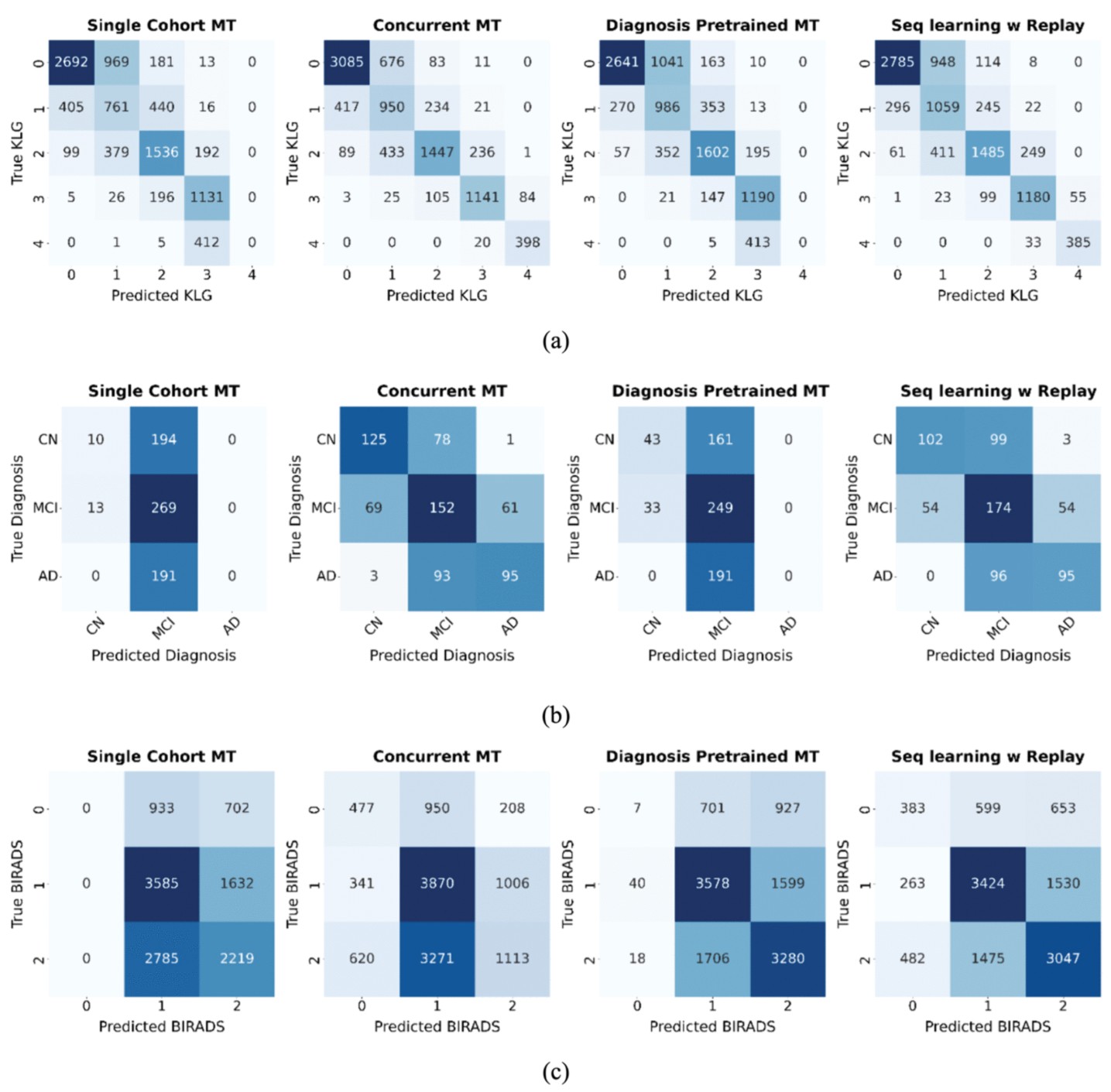

**Fig 4. Confusion matrices for the diagnosis task across different multitask learning approaches. (a)** Knee OA KLG prediction, **(b)** Alzheimer's diagnosis, and **(c)** Breast Cancer BI-RADS prediction. The matrices show that methods like 'Diagnosis Pretrained MT' forget how to classify rare classes (e.g., KLG 4, AD, BI-RADS 0) after prognosis fine-tuning. In contrast, 'Seq learning w/ Replay' retains knowledge across all classes, demonstrating its effectiveness at mitigating catastrophic forgetting.

This predictable decay in the baseline models highlights the essential value of the replay mechanism. By continually refreshing knowledge from the original task, experience replay preserves diagnostic competence – including for rare classes, while the model learns the new prognosis objective. The consistent success of this strategy across three physiologically distinct conditions (musculoskeletal, neurological, and oncological) suggests its broad applicability for other progressive diseases where longitudinal data is limited.

This work improves upon prior multitask learning strategies by demonstrating that the Single-cohort MT approach, similar to that used by Tiulpin et al. [14] and Leung et al. [16], is suboptimal for knee OA prognosis. Similarly in breast cancer, this study extends the findings of Wu et al. [13], showing that pretraining on diagnosis BI-RADS labels benefits not just breast cancer detection but also long-term prognosis.

### 4.2. Limitations

Several limitations warrant discussion. First, our knee OA and AD datasets derive from standardized research protocols, potentially lacking the variability of routine clinical practice and therefore affecting generalizability to diverse care settings. Second, the inherent class imbalance (few progressors versus many stable patients) in progressive disease studies affects all conditions studied. While we employed balanced sampling and specific evaluation metrics (AUPRC) to mitigate this, the imbalance remains a challenge for model deployment.

Third, a practical consideration for the sequential learning with experience replay strategy is its reliance on access to the original diagnosis dataset, which may not always be available when models are shared. It is important to note, however, that this limitation applies only to the multitask fine-tuning stage. The initial performance gains from using a diagnosis-pretrained model for standard fine-tuning – our first key finding remains an applicable benefit of our work, even without access to the original data for replay.

Finally, while experience replay effectively mitigates catastrophic forgetting, it increases the computational cost of training compared to simple fine-tuning, as the model must process 'replay' batches alongside new data.

### 4.3. Clinical implications

From a clinical perspective, these findings are relevant. A single, unified model that supports both diagnosis and prognosis mirrors routine workflow, assessing current status and planning forward. Such a model can aid counseling by presenting a fuller view – from present severity to future risk and can support personalization. For example, flagging patients at high risk of rapid OA progression for earlier intervention, or using screening mammograms to guide breast-cancer surveillance intensity. Because the sequential learning strategy preserves diagnostic competence, the model remains a dependable diagnostic aid while adding prognostic value, critical for trust and clinical adoption. In addition, producing diagnosis and prognosis together enables cross-verification at the point of care: clinicians can check whether current findings align with predicted risk and more closely review discordant cases, which may support trust and uptake. We view this as a usability/trust benefit rather than a validated safety outcome. Future prospective studies are needed to quantify its impact on decision quality. Additional future work should test these models on larger, more heterogeneous datasets from diverse care settings to assess generalizability beyond the research cohorts used here.

### 4.4. Conclusion

In conclusion, our work demonstrates an effective approach to extracting more value from limited longitudinal datasets by leveraging larger diagnosis datasets for prognostic model training.

### Supporting information

**S1 Fig. Comparative multitask learning approaches.** Visualizations of (a) Single Cohort MT, (b) Diagnosis Pretrained MT, and (c) Concurrent MT.
(TIF)

**S1 Table. Patient Characteristics in Structural Progression OAI and MOST Cohorts.**
(DOCX)

**S2 Table. Patient Characteristics in Structural Incidence OAI and MOST Cohorts.**
(DOCX)

**S3 Table. Demographic Characteristics and Cognitive Status Distribution across the ADNI Prognosis Cohort.**
(DOCX)

**S4 Table. Breast cancer Patient distribution in the Prognosis Cohort.**
(DOCX)

**S5 Table. Comparison of AUROC performance for models trained on progression and incidence, using different initializations.**
(DOCX)

**S6 Table. Detailed AUROC analysis comparing model performance across cognitive status subgroups.**
(DOCX)

**S7 Table. Detailed metrics for various training strategies for progression and incidence prediction tasks on OAI and MOST Datasets.**
(DOCX)

**S8 Table. Detailed progression prediction performance (AUROC) across cognitive status subgroups for multitask methods.**
(DOCX)

## Acknowledgments

We gratefully acknowledge data provision from the Osteoarthritis Initiative (OAI), a public-private partnership (this manuscript does not necessarily reflect the opinions of OAI investigators); the Multicenter Osteoarthritis Study (MOST); and the Alzheimer's Disease Neuroimaging Initiative (ADNI). A full list of ADNI contributors is available at adni.loni.usc.edu. We thank the investigators of these studies for their contributions.

## Author contributions

**Conceptualization:** Haresh Rengaraj Rajamohan, Kyunghyun Cho, Cem M. Deniz.

**Data curation:** Haresh Rengaraj Rajamohan, Yanqi Xu, Weicheng Zhu, Krzysztof J. Geras.

**Formal analysis:** Haresh Rengaraj Rajamohan.

**Funding acquisition:** Cem M. Deniz.

**Investigation:** Haresh Rengaraj Rajamohan, Cem M. Deniz.

**Methodology:** Haresh Rengaraj Rajamohan, Richard Kijowski, Cem M. Deniz.

**Project administration:** Narges Razavian.

**Resources:** Haresh Rengaraj Rajamohan.

**Software:** Haresh Rengaraj Rajamohan.

**Supervision:** Richard Kijowski, Krzysztof J. Geras, Narges Razavian, Kyunghyun Cho, Cem M. Deniz.

**Validation:** Haresh Rengaraj Rajamohan.

**Visualization:** Haresh Rengaraj Rajamohan.

**Writing – original draft:** Haresh Rengaraj Rajamohan, Yanqi Xu, Weicheng Zhu, Cem M. Deniz.

**Writing – review & editing:** Haresh Rengaraj Rajamohan, Yanqi Xu, Weicheng Zhu, Richard Kijowski, Krzysztof J. Geras, Narges Razavian, Kyunghyun Cho, Cem M. Deniz.

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
