## [Decision Letter · Decision Letter 0]

10 Dec 2025

PONE-D-25-52607Robust Disease Prognosis via Diagnostic Knowledge Preservation: A Sequential Learning ApproachPLOS One

Dear Dr. Rajamohan,

Thank you for submitting your manuscript to PLOS ONE. After careful consideration, we feel that it has merit but does not fully meet PLOS ONE’s publication criteria as it currently stands. Therefore, we invite you to submit a revised version of the manuscript that addresses the points raised during the review process.

Please improve the language proficiency to improve flow and clear information. Provide IRB details.

We look forward to receiving your revised manuscript.

Kind regards,

Naveen Baskaran, MD

Academic Editor

PLOS One

Journal Requirements:

“We gratefully acknowledge data provision from the Osteoarthritis Initiative (OAI), an NIH526 funded public-private partnership (this manuscript does not necessarily reflect the opinions of OAI investigators or funders); the Multicenter Osteoarthritis Study (MOST), sponsored by the NIH/National Institute on Aging; and the Alzheimer's Disease Neuroimaging Initiative (ADNI). ADNI is funded by NIH Grant U01 AG024904, DOD ADNI (W81XWH-12-2-0012), and numerous public and private contributions, with a full list of contributors available at adni.loni.usc.edu.”

“This work was supported by the National Institutes of Health (NIH) under grant R01-AR074453”

“This work was supported by the National Institutes of Health (NIH) under grant R01-AR074453”

Reviewers' comments:

Reviewer's Responses to Questions

**Comments to the Author**

1. Is the manuscript technically sound, and do the data support the conclusions?

Reviewer #1: Yes

Reviewer #2: Yes

2. Has the statistical analysis been performed appropriately and rigorously? 

Reviewer #1: Yes

Reviewer #2: Yes

3. Have the authors made all data underlying the findings in their manuscript fully available?

Reviewer #1: Yes

Reviewer #2: Yes

4. Is the manuscript presented in an intelligible fashion and written in standard English?

Reviewer #1: Yes

Reviewer #2: Yes

5. Review Comments to the Author

Reviewer #1: This manuscript is already well written and provides important insights into robust disease prognosis via diagnostic knowledge preservation. I have a few suggestions that may help improve the manuscript:

1. Line 91: Please provide the IRB institution and IRB number for this study.

2. Line 112: The abbreviation for ADNI should be introduced here, not at line 159.

3. Lines 162–163: Since the abbreviations for Cognitive Normal (CN), Mild Cognitive Impairment (MCI), and Alzheimer’s Disease (AD) have already been defined earlier, please use the abbreviations directly in this section.

4. Discussion: The manuscript would benefit from a dedicated Limitations section, as several inherent limitations appear to be present.

5. A thorough English language review is recommended to further improve clarity, flow, and readability.

Reviewer #2: A very convincing paper with sound findings & elaboration on the significance of the findings. A more recent literature would strengthened the juatification s of the findings. Critical argumenta are still lacking on some parts of discussions.

6. PLOS authors have the option to publish the peer review history of their article (what does this mean?). If published, this will include your full peer review and any attached files.

Reviewer #1: No

Reviewer #2: No

---

## [Author Response · Author response to Decision Letter 1]

6 Jan 2026

Dear Dr. Baskaran and Reviewers,

We thank the Academic Editor and the Reviewers for their thoughtful comments and constructive feedback on our manuscript. We are pleased that the reviewers found the study technically sound.

We have revised the manuscript to address the specific points raised, particularly regarding the inclusion of specific IRB details, the clarification of abbreviations, and the addition of a dedicated Limitations section. Our point-by-point responses are detailed below.

Reviewer #1

Comment: This manuscript is already well written and provides important insights into robust disease prognosis via diagnostic knowledge preservation

Response: We thank Reviewer 1 for their positive assessment and we appreciate their helpful feedback.

Comment 1: Line 91: Please provide the IRB institution and IRB number for this study.

Response: We thank the reviewer for this important request. We have updated the Ethics Statement (Section 2.1) to explicitly state that this is a retrospective analysis of fully de-identified data which did not require a separate IRB approval. We have added the available IRB approval numbers, ClinicalTrials.gov identifiers for the public datasets (OAI, MOST,ADNI) and the specific IRB number for the NYU breast cancer dataset.

Changes in the manuscript: “This retrospective study involved the analysis of fully de-identified data from three publicly available research datasets (OAI, MOST, and ADNI) and one institutional dataset (NYU Langone Health). The Osteoarthritis Initiative (OAI) (ClinicalTrials.gov identifier: NCT00080171) was approved by the Institutional Review Boards (IRB) at the UCSF Coordinating Center (approval #10-00532) and all clinical sites, including Memorial Hospital of Rhode Island, Ohio State University, University of Pittsburgh, and University of Maryland/Johns Hopkins University. Ethical approval for the Multicenter Osteoarthritis Study (MOST) was obtained from the IRBs at Boston University (H-32956), University of Alabama at Birmingham (IRB-000329007), University of California San Francisco (301480), and University of Iowa (201511711); approval for secondary data analysis was granted by the University of Florida (IRB202201899). Data for the Alzheimer’s Disease Neuroimaging Initiative (ADNI ClinicalTrials.gov identifier: NCT00106899) were obtained in accordance with the Declaration of Helsinki and approved by the IRBs of all participating sites. All participants in these public cohorts provided written informed consent. The Breast Cancer dataset was sourced from New York University Langone Health under a protocol approved by the NYU Langone Health IRB (IRB00010481, S18-00712). This dataset consisted of fully de-identified data and was approved for retrospective analysis with a waiver of informed consent, in compliance with HIPAA regulations. As this study involved only the retrospective analysis of fully de-identified data, it did not require new participant recruitment or additional IRB approval beyond the existing protocols cited above.“

Comment 2: Line 112: The abbreviation for ADNI should be introduced here, not at line 159.

Response: We have corrected this oversight. The abbreviation "(ADNI)" is now introduced at the first mention of the "Alzheimer’s Disease Neuroimaging Initiative" in the text (Section 2.1).

Comment 3: Lines 162–163: Since the abbreviations for Cognitive Normal (CN), Mild Cognitive Impairment (MCI), and Alzheimer’s Disease (AD) have already been defined earlier, please use the abbreviations directly in this section.

Response: We have updated Section 2.2.2 to use the abbreviations CN, MCI, and AD directly, as they were defined previously.

Comment 4: Discussion: The manuscript would benefit from a dedicated Limitations section, as several inherent limitations appear to be present.

Response: We agree that a transparent discussion of limitations is vital. We have added a dedicated subsection 4.2. Limitations within the Discussion section and expanded the previously mentioned Limitations.

Changes in the manuscript: The updated limitations subsection is as follows:

“Several limitations warrant discussion. First, our knee OA and AD datasets derive from standardized research protocols, while the breast cancer dataset originates from a single institution. These data sources may lack the variability of routine clinical practice and therefore affect generalizability to diverse care settings. Second, the inherent class imbalance (few progressors versus many stable patients) in progressive disease studies affects all conditions studied. While we employed balanced sampling and specific evaluation metrics (AUPRC) to mitigate this, the imbalance remains a challenge for model deployment.

Third, a practical consideration for the sequential learning with experience replay strategy is its reliance on access to the original diagnosis dataset, which may not always be available when models are shared. It is important to note, however, that this limitation applies only to the multitask fine-tuning stage. The initial performance gains from using a diagnosis-pretrained model for standard fine-tuning - our first key finding remains an applicable benefit of our work, even without access to the original data for replay.

Finally, while experience replay effectively mitigates catastrophic forgetting, it increases the computational cost of training compared to simple fine-tuning, as the model must process 'replay' batches alongside new data.”

Comment 5: A thorough English language review is recommended to further improve clarity, flow, and readability.

Response: We have carefully reviewed the entire manuscript and edited the text to improve clarity, flow, and readability as requested. We have tightened the phrasing in the Abstract and Introduction to ensure that key technical concepts are defined clearly. We have also corrected minor grammatical inconsistencies, improved sentence transitions, and refined the vocabulary throughout the Methods and Results sections to ensure a professional academic tone.

Changes in the manuscript: Below are a few representative examples of these revisions:

Section

Original Text

Revised Text

Abstract

"...hindered by the lack of long-term data."

"...hindered by the scarcity of longitudinal data."

Introduction

"Progressive diseases pose significant challenges... Their gradual onset and potential for irreversible damage..."

"Progressive diseases present significant challenges... Due to their gradual onset and potential for irreversible damage..."

Methods

"All models were trained using the Adam optimizer [33]. A cosine annealing learning rate scheduler was used..."

"All models were trained using the Adam optimizer [33], coupled with a cosine annealing learning rate scheduler..."

Results

"...weights pretrained on diagnostic tasks with full diagnosis cohorts**,** yielded improvements..."

"...weights pretrained on diagnostic tasks with full diagnosis cohorts yielded improvements..."

Results

"This performance benefit on the external MOST dataset"

"Additionally, this performance benefit on the external MOST dataset"

Discussion

"...exhibited lower variance, indicating more stable performance."

"...exhibited lower variance, suggesting more stable performance."

Discussion

"...mirrors routine workflow, assess current status, then plan forward."

"...mirrors routine workflow: assessing current status and planning forward."

Reviewer #2:

Comment: A very convincing paper with sound findings & elaboration on the significance of the findings.

Response: We thank Reviewer 2 for their positive assessment and we appreciate their encouraging feedback.

Comment 1:A more recent literature would strengthen the justifications of the findings. Critical arguments are still lacking in some parts of discussions.

Response: We agree that connecting our findings to the latest developments in medical AI strengthens the manuscript. We have added targeted, up-to-date citations in both the Introduction and Discussion to contextualize our approach relative to recent work on diagnostic-scale datasets and foundation models. Specifically, we now cite recent OA-focused AI literature and dataset-driven trends across progressive diseases [23–25], and we reference recent reviews and examples of medical foundation models that leverage large-scale diagnostic pretraining for improved downstream performance under limited labeled data [26,27]. In the Discussion (Section 4.1), we additionally connect our empirical findings to the broader paradigm of foundation models and multimodal AI [24,40], and we explicitly argue that our results extend these ideas to longitudinal prognosis, where labels are scarce and temporal prediction is challenging. Finally, to strengthen critical argumentation, we added a dedicated Limitations subsection (Section 4.2) that discusses key practical trade-offs, including generalizability, class imbalance, reliance on access to diagnostic data for replay, and the additional computational cost of experience replay.

Changes in the manuscript:

Introduction:

“However, developing robust prognosis models faces a significant hurdle: the scarcity of longitudinal data required to track disease progression over time. This contrasts with diagnosis, where larger cross-sectional datasets are often more readily available. As a result, recent advances in medical imaging AI - across knee osteoarthritis, Alzheimer’s disease, and oncology have largely focused on improving diagnostic accuracy and disease severity assessment, reflecting the scale and availability of diagnostic datasets rather than long-horizon progression labels [23-25]. To mitigate this data scarcity, previous works have explored multitask learning (MTL) as additional regularization, where a single model is trained to perform several related tasks simultaneously, aiming to improve generalization and performance through shared representations [13, 14, 16, 21, 22]. For instance, MTL combining diagnosis and prognosis has shown benefits in OA [14, 16], and pretraining on BI-RADS has been shown to improve current cancer diagnosis performance [13]. Yet these MTL studies typically rely on relatively small prognosis datasets and do not fully capitalize on the abundance of diagnostic data, even as recent medical foundation-model approaches increasingly leverage large-scale diagnostic pretraining with limited labeled data [26, 27]. This study therefore hypothesizes that sequential learning - diagnosis pretraining on large, unbiased datasets followed by prognostic training, achieves a favorable balance of prognostic and diagnostic performance.”

Discussion

“Recent work on generalist and multimodal medical AI suggests that representations learned from large-scale diagnostic data can serve as effective foundations for a wide range of downstream diagnostic and classification tasks, particularly in settings with limited labeled data [24,40]. Our results provide empirical evidence that this paradigm extends specifically to disease prognosis, where longitudinal labels are especially scarce and difficult to collect. By leveraging large, cross-sectional diagnostic cohorts, our approach improves prognostic discrimination and stability while preserving diagnostic competence, addressing a key gap between foundation-model principles and real-world longitudinal prediction tasks. ”

New Papers Cited:

[23] Harkey MS, Costello KE, Mehta B, Wen C, Malfait AM, Madry H, et al. Artificial intelligence in osteoarthritis research: summary of the 2025 OARSI pre-congress workshop. Osteoarthritis and Cartilage Open. 2025;100687.

[24] Moor M, Banerjee O, Abad ZS, Krumholz HM, Leskovec J, Topol EJ, et al. Foundation models for generalist medical artificial intelligence. Nature. 2023 Apr 13;616(7956):259-65.

[25] Weiner MW, Kanoria S, Miller MJ, Aisen PS, Beckett LA, Conti C, Diaz A, Flenniken D, Green RC, Harvey DJ, Jack Jr CR. Overview of Alzheimer's Disease Neuroimaging Initiative and future clinical trials. Alzheimer's & Dementia. 2025 Jan;21(1):e14321.

[26] D’Antonoli TA, Bluethgen C, Cuocolo R, Klontzas ME, Ponsiglione A, Kocak B. Foundation models for radiology: fundamentals, applications, opportunities, challenges, risks, and prospects. Diagnostic and Interventional Radiology. 2025.

[27] Ma, DongAo, et al. "A fully open AI foundation model applied to chest radiography." Nature (2025): 1-11.

[40] Acosta JN, Falcone GJ, Rajpurkar P, Topol EJ. Multimodal biomedical AI. Nature medicine. 2022 Sep;28(9):1773-84.

Further, the updated Limitations subsection is as follows:

“Several limitations warrant discussion. First, our knee OA and AD datasets derive from standardized research protocols, while the breast cancer dataset originates from a single institution. These data sources may lack the variability of routine clinical practice and therefore affect generalizability to diverse care settings. Second, the inherent class imbalance (few progressors versus many stable patients) in progressive disease studies affects all conditions studied. While we employed balanced sampling and specific evaluation metrics (AUPRC) to mitigate this, the imbalance remains a challenge for model deployment.

Third, a practical consideration for the sequential learning with experience replay strategy is its reliance on access to the original diagnosis dataset, which may not always be available when models are shared. It is important to note, however, that this limitation applies only to the multitask fine-tuning stage. The initial performance gains from using a diagnosis-pretrained model for standard fine-tuning - our first key finding remains an applicable benefit of our work, even without access to the original data for replay.

Finally, while experience replay effectively mitigates catastrophic forgetting, it increases the computational cost of training compared to simple fine-tuning, as the model must process 'replay' batches alongside new data.”

---

## [Editor Report · Decision Letter 1]

23 Feb 2026

Robust disease prognosis via diagnostic knowledge preservation: A sequential learning approach

PONE-D-25-52607R1

Dear Dr. Rajamohan,

We’re pleased to inform you that your manuscript has been judged scientifically suitable for publication and will be formally accepted for publication once it meets all outstanding technical requirements.

Kind regards,

Edward Hoffer, MD

Academic Editor

PLOS One
---

## [Editor Report · Acceptance letter]

PONE-D-25-52607R1

PLOS One

Dear Dr. Rajamohan,

I'm pleased to inform you that your manuscript has been deemed suitable for publication in PLOS One. Congratulations! Your manuscript is now being handed over to our production team.

Kind regards,

on behalf of

Dr. Edward Hoffer

Academic Editor

PLOS One